# Volatiles and Antifungal-Antibacterial-Antiviral Activity of South African *Salvia* spp. Essential Oils Cultivated in Uniform Conditions

**DOI:** 10.3390/molecules26092826

**Published:** 2021-05-10

**Authors:** Basma Najar, Giulia Mecacci, Valeria Nardi, Claudio Cervelli, Simona Nardoni, Francesca Mancianti, Valentina Virginia Ebani, Simone Giannecchini, Luisa Pistelli

**Affiliations:** 1Department of Pharmacy, University of Pisa, Via Bonanno Pisano 33, 56126 Pisa, Italy; g.mecacci@studenti.unipi.it (G.M.); nardivaleria12@gmail.com (V.N.); luisa.pistelli@unipi.it (L.P.); 2Council for Agricultural Research and Economics (CREA), Corso Inglesi 508, 18038 Sanremo, Italy; claudio.cervelli@crea.gov.it; 3Department of Veterinary Sciences, University of Pisa, Viale delle Piagge 2, 56124 Pisa, Italy; simona.nardoni@unipi.it (S.N.); francesca.mancianti@unipi.it (F.M.); valentina.virginia.ebani@unipi.it (V.V.E.); 4Interdepartmental Research Center “Nutraceutical and Food for Health”, University of Pisa, Via del Borghetto, 80, 56126 Pisa, Italy; 5Department of Experimental and Clinical Medicine, University of Florence, Viale Morgagni 48, 50134 Florence, Italy; simone.giannecchini@unifi.it

**Keywords:** *Salvia aurea*, *Salvia dentata*, *Salvia scabra*, dermatophytes, *Aspergillus*, *Fusarium solani*, *Staphylococcus aureus*, *Staphylococcus pseudointermedius*, influenza type A H1N1 virus

## Abstract

Spontaneous emissions of *S. dentata* Aiton and *S. scabra* Thunb., as well as the essential oil (EO) composition of the cited species, together with *S. aurea* L., were investigated. The chemical profile of the first two species is reported here for the first time. Moreover, in vitro tests were performed to evaluate the antifungal activity of these EOs on *Trichophyton mentagrophytes*, *Microsporum canis*, *Aspergillus flavus*, *Aspergillus niger*, and *Fusarium solani*. Secondly, the EO antibacterial activity against *Escherichia coli,* *Staphylococcus aureus*, and *Staphylococcus pseudointermedius* was examined, and their antiviral efficacy against the H1N1 influenza virus was assessed. Leaf volatile organic compounds (VOCs), as well as the EOs obtained from the arial part of *Salvia* *scabra*, were characterized by a high percentage of sesquiterpene hydrocarbons (97.8% and 76.6%, respectively), mostly represented by an equal amount of germacrene D (32.8% and 32.7%, respectively). Both leaf and flower spontaneous emissions of *S. dentata*, as well as the EO composition, showed a prevalence of monoterpenes divided into a more or less equal amount of hydrocarbon and oxygenated compounds. Interestingly, its EO had a non-negligible percentage of oxygenated sesquiterpenes (29.5%). *S. aurea* EO, on the contrary, was rich in sesquiterpenes, both hydrocarbons and oxygenated compounds (41.5% and 33.5%, respectively). *S. dentata* EO showed good efficacy (Minimal Inhibitory Concentration (MIC): 0.5%) against *M. canis*. The tested EOs were not active against *E. coli* and *S. aureus*, whereas a low inhibition of *S. dentata* EO was observed on *S. pseudointermedius* (MIC = 10%). Once again, *S. dentata* EO showed a very good H1N1 inhibition; contrariwise, *S. aurea* EO was completely inactive against this virus. The low quantity of *S. scabra* EO made it impossible to test its biological activity. *S. dentata* EO exhibited interesting new perspectives for medicinal and industrial uses.

## 1. Introduction

Throughout the world, thousands of people are affected by dermatophyte infections which constitute the most common skin diseases. These infections, especially caused by fungal pathogens belonging to *Trichophyton* and *Microsporum* species [1], are lead to the fourth highest incidence of disease when compared to hundreds of different illnesses and injuries globally [2]. On the other hand, *Staphylococcus aureus* is also considered the most common bacterial cause of skin problems and soft-tissue infections, as well as nosocomial bacteremia, in America and Europe [3], which colonizes in 20–30% of the human population [4], as well as livestock and domestic animals [5,6]. Another bacterium found in the nasal flora and skin of healthy dogs and cats is *S. pseudointermedius*. This bacterium occasionally infects human skin and soft tissue, leading to pneumonia, brain abscesses, and bacteremia [7]. *S. aureus* is considered a leading cause of foodborne disease around the globe [8]. *Escherichia coli* is another bacterium frequently involved in human and animal infections causing diseases with different degrees of severity [9].

Furthermore, fungi, especially the *Aspergillus* genus, are considered harmful to human health and a common cause of aspergillosis, a severe opportunistic infection [10,11]. Moreover, they are responsible for the contamination of foodstuffs. *Aspergillus niger* is, in fact, responsible for fruit spoilage, while strains of *A. flavus* produce aflatoxins, a potent hepatocarcinogenic agent in animals and humans [12,13], whereas *Fusarium solani* can produce mycotoxins.

In addition to this, humans must still face virus infections, especially influenza virus type A (IVA), a respiratory pathogen known to cause the flu pandemic (cdc.gov/flu/about/viruses/index.htm, accessed on 15 April 2021).

The investigation of new alternatives is of high priority, and the attentiveness of the scientific world toward the potential of using safe and effective molecules from natural origins is emerging. Thus, the interest in essential oil (EO), generally regarded as safe, is growing [14]. These oils are noted for their uses, which vary from food products to pharmaceuticals, and they represent one of the most promising approaches to fight infectious bacterial and viral microorganisms and to control antibiotic resistance [15]. Furthermore, their antibacterial, antifungal, and antiviral activities are well documented [16,17,18,19].

The *Salvia* genus includes about 30 species that grow in South Africa and, despite their use in the folk medicine, there are still only few studies proving their effective biological activity, which were almost exclusively dedicated to in vitro experiments [20]. Kamatou and Fisher were the only authors who investigated the South African *Salvia* EO, and only six species were the subject of their publications [21,22,23]. In order to contribute to the knowledge on the effect of new EOs, this work provides new insights into the chemical composition of three EOs from South African *Salvia* spp. grown in Italy, *S. aurea* L., *S. dentata* Aiton, and *S. scabra* Thunb., as well as their in vitro antimicrobial and antiviral properties.

## 2. Results and Discussion

### 2.1. Aroma Profile and EO Analyses

#### 2.1.1. Volatile Organ Compound Analysis

The aromatic profiles of the flowers and leaves of *S. dentata* and *S. scabra* are shown in Table 1, and their respective chromatograms are presented in Appendix A–d (Appendix A). Oxygenated monoterpenes were the main constituents in the flowers of both *S. dentata* and *S. scabra*, (57.9% and 53.1%, respectively), with three compounds found in common: 1,8-cineole (24.7% in *S. dentata* and 14.3% in *S. scabra*), camphor (30.4% in *S. dentata* and 37.0% in *S. scabra*), and bornyl acetate (0.4% in *S. dentata* and 6.8% in *S. scabra*). Interestingly, there was an absence of monoterpene hydrocarbons in the flower VOCs of *S. scabra*, while they represented 40.4% of volatile flower emissions of *S. dentata*, mainly represented by camphene (15.0%), α-pinene (10.7%), and β-pinene (6.4%). *S. scabra* flowers were characterized by a significant percentage of diterpene hydrocarbons (20.4%), of which cembrene (14.4%) and isopimara-9(11),15-diene (6.0%) showed the highest amount, together with oxygenated sesquiterpenes (13.8%), with nuciferol acetate (13.0%) as the most abundant compound.

The spontaneous emission of *S. scabra* leaves was almost exclusively characterized by sesquiterpene hydrocarbons (97.8%). This class was mostly represented by germacrene D (32.8%), β-caryophyllene (18.4%), α-copaene (13.5%), and γ-elemene (6.6%). On the contrary, monoterpenes predominated in the VOC leaves of *S. dentata* (oxygenated monoterpenes (47.5%) and monoterpene hydrocarbons (43.8%)). Camphor (22.4%), 1,8-cineole (16.3%), and bornyl acetate (6.8%) were the major OM constituents, while camphene (19.8%), α-pinene (11.8%) and β-pinene (4.9%) were the most abundant MHs. All of these compounds were also present in flowers of the same species. In the leaves of both sages, some common compounds were found, present in extremely variable percentages: myrcene (1.8% in *S. dentata* vs. 0.2% in *S. scabra*), camphor (22.4% in *S. dentata* vs. 0.5% in *S. scabra*), α-copaene (0.3% in *S. dentata* vs. 13.5% in *S. scabra*), and β-caryophyllene (1.6% in *S. dentata* vs. 18.4% in *S. scabra*).

There are still few studies concerning the flavor profile of South African sage species. In fact, according to the best of our knowledge, no work has reported the spontaneous emissions of *S. dentata* and *S. scabra*. The articles present in the literature, so far, focused on other species. Ascrizzi [26], studying the VOC composition of *S. aurea* and *S. aurita* L.f., evidenced the prevalence of monoterpene hydrocarbons (93.4%) and sesquiterpene hydrocarbons (44.4%), respectively. These results disagreed with our findings in *S. dentata*, in which oxygenated monoterpenes prevailed (57.9% in the flowers and 47.5% in the leaves), although a considerable percentage of monoterpene hydrocarbons was observed (greater than 40.0% both in the leaves and in the flowers). On the contrary, the volatile organic compounds of *S. scabra* leaves (97.8% sesquiterpene hydrocarbons) had a similar trend to *S. aurita* [26]. Myrcene and β-caryophyllene were the most abundant compounds in *S. aurea* (89.5%) and *S. aurita* (17.8%), respectively. Myrcene was present in a reduced percentage only in the leaves of *S. dentata*, while β-caryophyllene was more abundant in *S. scabra*, even though the prevalent compound was germacrene D.

Both leaves and flowers of *S. uliginosa* Benth., studied by Giuliani et al. [27], were characterized by a high percentage of sesquiterpene hydrocarbons (90.1% and 92.18%, respectively). The most abundant compounds in flowers were biciclogermacrene (31.3%) and β-caryophyllene (25.6%). In the leaves, the above-cited compounds were also present (16.9% and 12.9%, respectively) even though γ-muurolene (31.4%) was the predominant constituent. Analyzing the *S. scabra* leaf aromatic profile, only β-caryophyllene was present in a high percentage (18.4%), while γ-muurolene was present in a reduced amount (0.1%); biciclogermacrene was completely absent.

#### 2.1.2. Essential Oil Analyses

The composition of the investigated EOs from the three sages is shown in Table 2, and their representative chromatograms are presented in Appendix A (Appendix A). Overall, 125 compounds were identified, accounting for at least 98.1% of the whole essential oil composition (Table 2). Out of these constituents, only 15 were found in common among the three species: α- and β-pinene, myrcene, *p*-cymene, limonene, δ-3-carene, 1,8-cineol, terpinolene, camphor, α-copaene, β-caryophyllene, α-humulene, δ-cadinene, viridiflorol, and pentacosane. The extraction yield of *S. scabra* oil was very low (0.10%), while that for *S. aurea* and *S. dentata* was quite high (1.01% and 1.53%, respectively). Sesquiterpenes hydrocarbons and oxygenated sesquiterpenes were the predominant classes in both *S. scabra* (76.6% and 15.8%, respectively) and *S. aurea* (41.5% and 33.5%, respectively), which boasted of a good percentage of MHs (17.0%).

On the contrary, *S. dentata* EO was characterized by similar amounts of oxygenated monoterpenes (35.1%) and monoterpene hydrocarbons (32.4%). The predominant compounds in *S. aurea* were β-caryophyllene (12.5%), epi-α-cadinol (10.2%), δ-cadinene (7.8%), and δ-3-carene (7.8%), while germacrene D (32.7%), β-caryophyllene (8.4%), germacrene B (7.8%), and α-copaene (6.5%) were the most abundant constituents in *S. scabra*. Despite the prevalence of monoterpenes, viridiflorol, an oxygenated sesquiterpene, was the predominant constituent in *S. dentata* (27.7%), followed by camphor (23.0%), α-pinene (10.2%), and camphene (10.0%).

Comparing the current results of the *S. aurea* EO with those reported in the literature, some substantial differences can be noted. First of all, the main compounds changed depending on the use of fresh or dry material. In fact, fresh leaves of *S. aurea* (native from South Africa but grown in Italy), analyzed by Serrato-Valenti [28], pointed out presented (34.7%), δ-3-carene (16.5%), and camphene (8.3%) as the predominant constituents. The oil extracted herein from dry material showed β-caryophyllene (12.5%) and epi-α-cadinol (10.2%) as the most abundant components, although a considerable percentage of δ-3-carene (7.8%) was also noted. Moreover, both camphor and camphene were present, but in extremely low percentages (0.2% and 0.1%, respectively). The dried aerial parts of native South African *S. aurea* EO was characterized by monoterpene hydrocarbons (35.6%), with myrcene as the most abundant constituent (11.5%) [29]. These were in total disagreement with our findings, and the myrcene value did not exceed 1.0%.

The same authors [20], 2 years later, showed how the composition, yield, and biological activity of the EO from South African *S. aurea* changed depending on the harvest season. The highest essential oil yield was obtained in September and October. α-Eudesmol (12.9% in December), β-eudesmol (12.7% in December), myrcene (11.5% in November), and α-pinene (11.9% in July) were the main components. The *S. aurea* sample studied herein was collected between May and June 2017, and only α-pinene (1.3%) and myrcene (1.0%) were present in lower percentages. Still analyzing the South African species, van Vuuren [30] confirmed the dominance of β-eudesmol (14.5%) and α-eudesmol (13.5%) in *S. aurea* EO. Other compounds were also present in a high amount such as α-pinene (8.6%), δ-3-carene (7.4%), β-caryophyllene (5.9%), limonene (5.3%), β-phellandrene (5.3%), γ-eudesmol (4.3%), epimanool (3.9%), viridiflorol (3.6%), β-pinene (3.4%), and myrcene (3.3%). Only δ-3-carene was present in almost the same amount in the studied oil. The remaining constituents were present in a lesser percentage or completely absent, such as β-eudesmol, α-eudesmol, β-phellandrene, γ-eudesmol, and epimanool.

The same species collected in the Western Cape region of South Africa in the vegetative stage was the subject of a more recent work [31]. The authors registered a yield of EO very low in comparison with that found herein (0.25% vs. 1.01%) but in agreement with the amount found by Kamatou [29]. They also observed that β-eudesmol (12.3%), α-eudesmol (12.4%), terpinene-4-ol (10.1%), and T-cadinol (7.6%) were the majority. This was in contrast with the results of this study, where the EO was eudesmol-free and terpinene-4-ol was of a very less percentage (0.2%).

To the best of our knowledge, the chemical composition of *S. dentata* and *S. scabra* EOs has never been reported in the literature. However, numerous works investigated the chemical composition of EOs from different South African *Salvia* species. Fisher [22] studied the chemical profile of *S. disermas* L., *S. dolomitica* Codd, and *S. namaensis* Schinz EOs. Linalyl acetate (34.5%) prevailed in the first species, but it was completely absent in all the sages analyzed herein. The second evidenced important percentages of 1,8-cineole (17.6%), β-caryophyllene (17.4%), and limonene (9.7%); all these compounds were found in reduced amounts in our work. The *S. namaensis* EO, instead, presented camphor (33.5%), camphene (14.7%), α-pinene (9.3%), 1,8-cineole (8.2%), and bornyl acetate (6.8%) as predominant compounds [22]. All these compounds were observed in *S. dentata*, even though the main component was viridiflorol (27.7%).

Kamatou [29] examined the EO composition of *S. africana-caerulea* L. and *S. lanceolata* Lam. and found oxygenated sesquiterpenes as the predominant constituents (58.7% and 47.9%, respectively), mostly represented by spatulenol (29.1% and 18.3%, respectively) and caryophyllene oxide (14.0–15.0%). This chemical class, although present in all the examined species in consistent percentages (33.5% in *S. aurea*, 29.5% in *S. dentata*, and 15.8% in *S. scabra*), did not predominate in any of them. The highest percentage of caryophyllene oxide was found in *S. aurea* (3.6%) and *S. scabra* (3.4%), while its level in *S. dentata* did not exceed 0.2%. A small percentage of spatulenol, on the other hand, was present only in *S. scabra*.

Contrary to previous results, *S. chameleaeagnea* Berg. stood out for the prevalence of oxygenated monoterpenes (42.8%), particularly 1,8-cineole (40.5%) [29]. The same trend was observed in *S. dentata*, even though camphor prevailed and 1,8-cineole was present at only 4.1% of the identified fraction. Other researchers [27] investigated the EO profile of another South African species, *S. uliginosa* Benth. and evidenced the dominance of bicyclogermacreme (16.3%), germacrene D (14.8%), and spathulenol (12.7%). Here, only *S. scabra* EO showed a good amount of germacrene D (32.7%). The other two constituents were almost absent in all three studied species.

*S. somalensis* Vatke and *S. dolomitica* were characterized by monoterpenes (73.1% in *S. dolomitica* and 67.8% in *S*. *somalensis*) [32] as observed in *S. dentata* EO (67.5%). 1,8-Cineole was the main constituent in *S*. *dolomitica* (18.9%), while bornyl acetate (16.1%) and camphor (12.5%) showed the highest percentages in *S*. *somalensis*. *β*-Caryophyllene (13.1% in *S*. *dolomitica*) and *δ*-cadinene (6.4% in *S. somalensis*) were the most representative compounds of sesquiterpene hydrocarbons [32]. All these compounds were present in different relative amounts in the investigated sages except for bornyl acetate, which was absent in *S. aurea*. Viridiflorol, the main compound of *S. dentata*, was also the main constituent in both *S. africana-caerulea* (36.7%) and *S. chamelaeagnea* (32.5%) [30].

### 2.2. Antimicrobial and Antiviral Activities

The examined EOs showed variable antimycotic degree toward the tested fungi (Table 3). The microdilution test showed that only dermatophytes were sensitive to the sage Eos, particularly *S. dentata*. The low quantity of *S. scabra* EO made it impossible to test its antimicrobial and antiviral activity. Noteworthily, *S. dentata* EO demonstrated a good antifungal action on *M. canis* (MIC = 0.5%). The tested oils were not active against *E. coli* and *S. aureus*, whereas a low inhibition efficacy of *Salvia dentata* EO was observed on *S. pseudointermedius* (MIC = 10%). Regarding the antiviral activity, only *S. dentata* EO showed very good H1N1 inhibition. On the contrary, *S. aurea* was completely inactive against this virus (Table 4).

As far as we know, no report is present in the literature on the biological activity of *S. dentata* EO. Only *S. aurea* EO was the subject of a few studies for its biological activity. The first one dates back to 1998 when Bisio and collaborators investigated its antimicrobial activity [33]. The authors tested this oil on 13 microorganisms. They found a nonsignificant effect on Gram-positive bacteria, especially *S. aureus*. These results were confirmed in this work. Later on, Russo tested the oil firstly on human melanoma cells (M14, A2058 and A375) [34] and then on prostate cancer [35]. In both works, the author affirmed the inhibition of growth and an apoptotic effect on all the tested cells.

Kamatou and collaborators [36] investigated other South African species and noted a poor antimicrobial activity of the three sage EOs. Later, van Vuuren [30] demonstrated the highest activity of *S. africana-caerulea* EO against the *Brevibacillus agri* foot-odor causing bacterium. This oil was characterized by viridiflorol (36.7%) and limonene (25.7%).

Viridiflorol was also found as the major constituent in Algerian *S. algeriens* Desf. and Iranian *S. sclareopsis* flower (71.1%) and leaf EOs (23.47%). Antifungal activity against *Alternia solani* and *Fusarium oxysporum* was noted using the flower of Algerian sage even though the best effectiveness was observed using the leaf EO (rich in benzaldehyde, eugenol, and phenylethyl) [37]. The* S. sclareopsis* leaf EO, however, evidenced a high antioxidant activity [38]. These results confirmed those found here, where the *S. dentata* EO, i.e., that with the highest amount of viridiflorol, was completely inactive on the tested *Fusarium* spp.

The investigation on Tunisian sage (S*. officinalis* L.), where camphor was one of main compounds (25.14%) as in our *S. dentata* EO, reported an interesting activity of this oil against *S. aureus* [39]. Their results disagreed with ours because *S. dentata* EO showed only slight activity. The observed activity of *S. dentata* EO could be due to the presence of both viridiflorol and camphor. Da Silva [40], in fact, demonstrated that viridiflorol, even though in a low amount, was more effective against *S. aureus*, while Gilabert [41] noted an inhibition of about 40% of the growth of human pathogenic bacteria using the same compound (viridiflorol) at 50 µg/mL. Trevizan [42] also showed the in vitro efficacy of the cited component but on *Mycobacterium tuberculosis* (MIC = 190.0 μg/mL), and they compared the in vivo anti-inflammatory activity to dexamethasone. This compound was also noted for its potent acetylcholinesterase inhibition [43]. A moderate antioxidant activity of *Ferula vesceritensis*, where viridiflorol was one of the main compounds (13.4%), was reported by Benchabne et al. [44].

Observing the results of antiviral activity once again, only the *S. dentata* EO evidenced very good action on H1N1 virus (Table 5). This oil presented a fair percentage of β-pinene, (3.2%), a compound known to have good anti-HSV activity, as stated by Orhan [45] and Astani [46], whereas, α-pinene reduced the infectivity of HSV-1 by >96% [47]. The cited constituent was present at a highest percentage in the active sage oil (10.2%). Falang carried out an in silico investigation of the plant constituents of some Nigerian medicinal species [48]. The authors showed that the phytochemical constituents of the selected plants had better binding affinity to several Covid-19 viral target proteins, testing the *S. officinalis* EO compounds borneol, camphor, and pinene. Thus, this plant was selected among nine others to proceed with in vitro studies. These constituents were almost exclusive to *S. dentata* essential oil and could be responsible for the antiviral action of this oil.

Kamatou [36] assessed the antimalarial activity of three South African sage species and pointed out that the best antimalarial and anti-inflammatory activity was shown by *S. runcinate* EO, where β-caryophyllene (10.5%) was one of main compounds, as in *S. aurea*. The *S. aurea* EO used herein was ineffective on the tested virus. This could be explained, on one hand, by the fact that both malaria and influenza virus evidenced different sensitivity to this oil and, on the other hand, by the fact that the antimalaria activity may have been due to a synergic effect of other compounds [49,50].

β-Caryophyllene, a compound shared by different *Salvia* species, also suppressed HSV multiplication by more than 90% [51]. An investigation on *Mosla dianthera* EO, which showed a comparable amount of β-caryophyllene, confirmed its safe and effective therapeutic ability for the treatment of influenza and subsequent viral pneumonia [52]. Recently, Dunkic [53] demonstrated that, in addition to β-caryophyllene content, germacrene D might play an important role in the antiphytoviral activity. These results were in complete disagreement with those found in this work, where, although *S. aurea* contained large amounts of β-caryophyllene, it was completely inactive on the tested virus. Others compounds were found to inhibit virus replication with a dosage below the cytotoxic level, such as terpinen-4-ol, terpinolene, and α-terpineol [54]. All of these compounds were present in the analyzed sage samples, except for α-terpineol, which was present only in *S. dentata* EO, in a very low amount; however, this could explain the activity of this latter EO on H1N1 virus.

## 3. Materials and Methods

### 3.1. Origin and Cultivation Method of the Plant Material

The native South African *Salvia* spp. (Appendix A) are part of the aromatic plant collection of the Research Center for Horticulture of CREA in Sanremo (Italy). The seeds were purchased from specialized companies during a seed sale of plants from the African flora (Silver Hill—PO Box 53108, Kenilworth, 7745 Cape Town, South Africa and B & T World Seeds—Paguignan, 34210 Aigues Vives, France). The plants were grown in pots under the same edaphic and climatic conditions. After clonal propagation, the plants grew in pots in the open air and were watered periodically. Flowering took place after 1 year. Plant samples were deposited in the herbarium of the Hanbury Botanical Gardens (La Mortola-Ventimiglia, Imperia, Italy). The correct identification of the plants was carried out by Claudio Cervelli. The aerial parts were harvested in the flowering period and dried at room temperature for 5 days.

### 3.2. Phytochemical Analysis

#### 3.2.1. Volatile Organ Compound and Essential Oil Analyses

The analysis of volatile organic compounds was performed on fresh plant using the solid-phase microextraction (SPME) method [55]. A sample of the fresh aerial parts of each *Salvia* sp. was placed separately in a glass jar, and then sealed with aluminum foil for 30 min (equilibration time) at room temperature (22 ± 1 °C). By the end, the fiber (PDMS, 100 µm) (St. Louis, MO, USA), previously preconditioned according to the manufacturer’s instructions, was exposed to the headspace for 15 min. Once sampling was finished, the fiber was withdrawn into the needle and transferred to the injection port of the GC–MS instrument, where thermal desorbing and component analysis took place. It was not possible to analyze the spontaneous emissions of *S. aurea* because the fresh plant was not available. For the essential oil extraction, the dried aerial parts of each plant were hydrodistilled using the Clevenger apparatus according to the European Pharmacopoeia (EDQM, 2017). The obtained essential oil was kept at a temperature of 4 °C and away from light sources until analysis. A diluted oil, in *n*-hexane by HPLC (at 5%), was injected into GC–MS.

#### 3.2.2. GC–MS Analyses

Gas chromatography–mass spectrometry (GC–MS) was used to determine VOC and EO components. The gas chromatograph used was an Agilent 7890B (Agilent Technologies Inc., Santa Clara, CA, USA). The mobile phase was represented by helium (He). The capillary column was an Agilent HP5-MS (Agilent Technologies Inc., Santa Clara, CA, USA), of 30 m length and 0.25 mm diameter. The stationary phase was linked to the internal surface of the column via covalent bonds and was stabilized by transverse bonds. The syringe was inserted into the gas chromatograph through the injector, allowing the adsorbent fiber to come out. The splitless method was used for injection; the injected sample was vaporized and transported to the carrier gas column. The temperature at the injector level was 220 °C. The separation column was contained in a thermostatic chamber, in which the starting temperature was 60 °C, increasing by 3 °C per minute up to 240 °C. The detector coupled to the gas chromatograph was an Agilent 5977B single-quadrupole mass spectrometer (Agilent Technologies Inc., Santa Clara, CA, USA), operating in full scan mode (1 scan/s), in the range 30–300 *m*/*z*.

The compounds were identified by comparing their retention times with those of pure reference samples and comparing their linear retention indices (LRIs), determined relative to a series of *n*-alkanes. The comparison was made by software with the constituents present in the commercial libraries NIST 2014 and Adams 2007 [56,57,58,59,60,61,62].

### 3.3. Antimicrobial Analyses

#### 3.3.1. Evaluation of Antifungal Activity

The EOs were tested on dermatophytes (*M. canis* and *T. mentagrophytes*), isolated from animal hair samples and potentially mycotoxin-producing molds (*A. niger*, *A. flavus*, and *F. solani*) isolated from environmental sources. The molds were maintained on Potato Dextrose Agar at −20 °C. The antimycotic activity of EOs was investigated using a microdilution test as recommended by Clinical and Laboratory Standards Institute M38A_2_ [60] (CLSI, 2008) and following the protocol described by Ebani [63], which evaluated the growth of the fungus in culture media added with scalar concentrations (*v*/*v*) at 5%, 4%, 3%, 2%, 1%, 0.25%, 0.2%, and 0.1%. All assays were performed in triplicate. Positive controls were achieved using itraconazole for dermatophytes and amphotericin B; the negative control was culture medium alone.

#### 3.3.2. Antibacterial Activity

The tests were executed using three canine clinical isolates, specifically, one *E.*
*coli* strain and one *S.*
*aureus* strain previously isolated from dogs with urinary tract infections, and one *S. pseudointermedius* strain isolated from a dog with otitis. The antibacterial activity of essential oils was tested using both the diffusion agar method (Kirby–Bauer) and the broth microdilution test.

##### Agar Disc Diffusion Method (Kirby–Bauer Technique)

The agar disc diffusion method was executed following the procedures described by Clinical and Laboratory Standards Institute [64] and with some modifications as previously described. Briefly, 9 cm diameter petri dishes containing Muller–Hinton medium were sown, through the use of a swab, with the bacterial strain in order to obtain uniform bacterial growth. Sterile cellulose 6 mm discs soaked in a solution (10% *v*/*v* in dimethyl sulfoxide (DMSO)) of each EO were added. The in vitro sensitivity of all bacterial strains to chloramphenicol was assayed using the same method, and the results were interpreted as indicated by the National Committee for Clinical Laboratory Standards [65]. The plates were then incubated for 24 h at 37 °C. The diameters of inhibition zones (IZs) were measured in millimeters, and the tests were performed in triplicate.

##### Minimum Inhibitory Concentration

The MIC value (minimum inhibitory concentration) was determined by the broth microdilution method following the protocol previously reported [63]. In brief, an EO stock solution was prepared by adding 40 µL of each oil to 360 µL of BHI (Brain Heart Infusion) broth. The test involved the preparation of a series of decreasing scalar dilutions (halved) of the antimicrobial agent, to which the same amount of BHI was added. In each well of a 96-well sterile microplate, 95 µL of BHI was added to 95 µL of the stock solution (10% solution). Then, 5 µL of each bacterial suspension was inoculated into each well. The test was performed in a total volume of 100 µL with final EO concentrations of 10% to 0.5%. The same assay was performed simultaneously for microorganism growth control (tested agents and media) and sterility control (tested oil and media). All tests were performed in triplicate, with chloramphenicol (Oxoid Ltd., Basingstoke, UK) as a positive control.

The microplates were incubated at 37 °C for 24 h. The MIC value was defined as the lowest concentration, expressed as mg/mL, of each EO for which microorganisms showed no visible growth.

### 3.4. Antiviral Activity

Madin-Darby canine kidney (MDCK) cells, propagated in modified Eagle’s medium (MEM; SIGMA, Milano, Italy) supplemented with 10% fetal bovine serum (FBS; SIGMA) and 1% penicillin/streptomycin (SIGMA), were used for the inhibitory viral plaque reduction assay (PRA). Briefly, six-well plates were seeded with 2.5 × 10^5^ cells in 3 mL of growth medium and kept overnight in incubators at 37 °C with 5% CO_2_. On the day of infection, after removal of the growth medium, cell monolayers at 80–90% confluence were infected with 100 mL of influenza virus H1N1 (human pandemic variant A/Firenze/05/2017 H1N1) with a multiplicity of infection (MOI) of 0.01 in the presence or absence (MEM with DMSO alone) of different concentration (from 0.1% to 0.0001%) of each EO diluted in DMSO in a final volume of 0.3 mL and incubated for 1 h at 37 °C with 5% CO_2_. Then, after a washing step with PBS 1×, the overlay medium composed of 0.5% Sea Plaque Agarose (Lonza, Basel, Switzerland) diluted in propagation medium supplemented with l-1-tosylamido-2-phenylethyl-chloromethyl-ketone-treated trypsin (2 mg/mL; Sigma, St. Louis, MO, USA) was added to each well. After 4 days of incubation at 37 °C, the monolayers were fixed with methanol (Carlo Erba Chemicals, Milan, Italy) and stained with 0.1% crystal violet (Carlo Erba Chemicals), and the viral titers were calculated on the basis of counting plaque-forming units (PFU). The percentage of PRA was calculated by dividing the average PFU of EO-treated samples by the average of untreated samples (viral positive control in the presence of DMSO alone): PRA = 100 − (PFU obtained with EOs at indicated dilution/PFU obtained with DMSO alone) × 100. All experiments were repeated at least twice.

## 4. Conclusions

The volatile emissions of *S. dentata* and *S. scabra*, as well as the EO composition of the cited species, together with *S. aurea* (South Africa sages), were investigated. The chemical profiles of the first two species were reported here for the first time. The EOs obtained in good amounts were tested for antifungal, antibacterial, and antiviral activity. It is worthy to note that the effect of *S. dentata* essential oil was startling and showed a fair to good activity on the tested pathogens. These plants were introduced in Italy for ornamental purposes by CREA-OF (Sanremo); however, according to these encouraging biological activities, the EO of *S. dentata* deserves deep investigation and could exhibit interesting new perspectives for medicinal and industrial uses. The *S. sclarea* EO presented compounds previously known for their good antiviral activity; therefore, the use of another extraction method would be interesting to increase its EO yield in order to verify this potential.

## Figures and Tables

**Table 1 molecules-26-02826-t001:** Aromatic profiles of *S. dentata* and *S. scabra* flowers and leaves.

					Flowers		Leaves	
	Compounds ^a^	Class	L.R.I ^exp^	L.R.I ^lit^	*S. dentata*	*S. scabra*	*S. dentata*	*S. s cabra*
					Relative percentage (%)			
1	Tricyclene	mh	927	921	0.1 ± 0.1		0.5 ± 0.0	
2	α-Thujene	mh	930	924	0.4 ± 0.0			
3	α-Pinene	mh	939	932	10.7 ± 0.2		11.8 ± 1.3	
4	Camphene	mh	954	946	15.0 ± 0.7		19.8 ± 0.9	
5	β-Pinene	mh	979	974	6.4 ± 0.0		4.9 ± 0.1	
6	Myrcene	mh	991	988			1.8 ± 0.2	0.2 ± 0.1
7	α-Phellandrene	mh	1003	1002	0.1 ± 0.1			
8	*p*-Mentha-1(7),8-diene	mh	1004	1003			0.1 ± 0.1	
9	(*Z*)-3-Hexenol acetate	nt	1005	1004				0.3 ± 0.1
10	*p*-Cymene	mh	1025	1020	0.3 ± 0.1		0.1 ± 0.2	
11	Limonene	mh	1029	1024	1.4 ± 0.5			0.6 ± 0.2
12	1,8-Cineole	om	1031	1026	24.7 ± 0.2	14.3 ± 0.6	16.3 ± 0.1	
13	*δ*-3-Carene	mh	1031	1021 *	4.0 ± 0.2		3.8 ± 0.3	
14	Butyl isovalerate	nt	1047	1048 *			1.5 ± 0.2	
15	γ-Terpinene	mh	1060	1054	0.8 ± 0.1		0.4 ± 0.1	
16	*cis*-Sabinene hydrate	om	1070	1065	0.4 ± 0.0		0.1 ± 0.2	
17	Terpinolene	mh	1089	1086	1.2 ± 0.0		0.6 ± 0.1	
18	*trans*-Sabinene hydrate	om	1098	1098	0.2 ± 0.1		0.1 ± 0.2	
19	Linalool	om	1099	1095			0.1 ± 0.2	
20	*cis*-Thujone	om	1102	1101	0.2 ± 0.1			
21	2-Methylbutyl isovalerate	nt	1107	1103			0.4 ± 0.0	
22	*n*-Amyl isovalerate	nt	1108	1108 *			1.9 ± 0.1	
23	*trans*-Thujone	om	1114	1112	0.2 ± 0.0		0.4 ± 0.0	
24	*allo*-Ocimene	om	1132	1128	0.3 ± 0.0		0.4 ± 0.0	
25	Camphor	om	1146	1141	30.4 ± 0.7	37.0 ± 0.9	22.4 ± 0.7	0.5 ± 0.5
26	Borneol	om	1169	1165	0.9 ± 0.1		0.4 ± 0.0	
27	4-Terpineol	om	1177	1174	0.2 ± 0.0		0.5 ± 0.0	
28	(*Z*)-3-Hexenyl butyrate	nt	1186	1184			0.4 ± 0.1	
29	Decanal	nt	1202	1198		1.6 ± 0.3		
30	(*Z*)-3-Hexenyl isovalerate	nt	1238	1235 *			0.1 ± 0.0	
31	Hexyl 3-methylbutanoate	nt	1244	1242 *			0.2 ± 0.0	
32	Bornyl acetate	om	1289	1284	0.4 ± 0.1	1.8 ± 0.4	6.8 ± 1.2	
33	2,2,4,4,6,8,8-Heptamethylnonane	nt	1322	1317 *		0.7 ± 1.0		
34	*δ*-Elemene	sh	1338	1335				0.3 ± 0.1
35	α-Copaene	sh	1377	1374			0.3 ± 0.0	13.5 ± 0.8
36	β-Patchoulene	sh	1381	1379				0.1 ± 0.1
37	β-Bourbonene	sh	1388	1387				0.8 ± 0.1
38	β-Cubebene	sh	1388	1387				0.9 ± 0.0
39	β-Elemene	sh	1391	1389				2.4 ± 0.5
40	*cis*-*α*-Bergamotene	sh	1413	1411				0.1 ± 0.1
41	β-Caryophyllene	sh	1419	1417	1.5 ± 0.2		1.6 ± 0.0	18.4 ± 0.3
42	β-Copaene	sh	1432	1430				5.2 ± 0.19
43	(+)-α-Barbatene	sh	1436	1437 *				1.5 ± 0.2
44	γ-Elemene	sh	1437	1434				6.6 ± 1.5
45	10,10-Dimethyl-2,6-dimethylenebicyclo[7.2.0]undecane	sh	1440	1440 *	0.1 ±0.1		0.1 ± 0.0	
46	Isogermacrene D	sh	1448	1446 ^$^				1. 9± 0.12
47	(*E*)-Geranylacetone	ac	1455	1453		1.6 ± 2.0		
48	α-Humulene	sh	1455	1452			0.1 ± 0.0	
49	*allo*-Aromadendrene	sh	1460	1458	0.1 ± 0.0		0.3 ± 0.0	0.2 ± 0.0
50	9-*epi*-(*E*)-Caryophyllene	sh	1466	1464				5.5 ± 0.2
51	*cis*-Muurola-4(14),5-diene	sh	1467	1465				0.7 ± 0.1
52	1-Dodecanol	nt	1473	1469		0.5 ± 0.3		
53	γ-Muurolene	sh	1480	1478				0.1 ± 0.0
54	Germacrene D	sh	1485	1484				32.8 ± 1.0
55	Valencene	sh	1492	1496				0.9 ± 0.6
56	γ-Amorphene	sh	1496	1495				2.2 ± 0.1
57	α-Muurolene	sh	1500	1500				0.1 ± 0.1
58	(*E,E*)-*α*-Farnesene	sh	1508	1505				1.0 ± 0.1
59	α-Chamigrene	sh	1508	1503				0.1 ± 0.0
60	Tridecanal	nt	1512	1509		0.1 ± 0.1		
61	*trans*-γ-Cadinene	sh	1514	1513				0.1 ± 0.0
62	1,2-Dihydrocuparene	sh	1521	1521 *				0.1 ± 0.0
63	*δ*-Cadinene	sh	1523	1522				1.3 ± 0.1
64	(*E*)-γ-Bisabolene	sh	1533	1529				0.8 ± 0.1
65	Germacrene B	sh	1561	1559				0.2 ± 0.0
66	*n*-Tridecan-1-ol	nt	1577	1570		0.5 ± 0.8		
67	Viridiflorol	os	1593	1592			0.8 ± 0.1	
68	Hedione	nt	1649	1650 *		1.1 ± 1.5		
69	2-Ethylhexyl octanoate	nt	1688	1688 *		3.2 ± 4.5		
70	*cis*-Valeranyl acetate	os	1817	1828 *		0.8 ± 1.1		
71	Nuciferol acetate	os	1837	1830		13.0 ± 4.0		
72	Isopimara-9(11),15-diene	dh	1906	1905		6.0 ± 1.5		
73	Cembrene	dh	1939	1937		14.4 ± 1.1		
74	*n*-Eicosane	nt	2000	2000			0.1 ± 0.1	
75	Isopropyl palmitate	nt	2026	2023 *		0.5 ± 0.7		
76	Hexacosane	nt	2600	2600			0.7 ± 1.0	
					**Flowers**		**Leaves**	
	Class of compounds				*S. dentata*	*S. scabra*	*S. dentata*	*S. scabra*
	Monoterpene hydrocarbons (mh)				40.4 ± 0.5		43.8 ± 3.1	0.8 ± 0.1
	Oxygenated monoterpenes (om)				57.9 ± 0.1	53.1 ± 0.7	47.5 ± 2.2	0.5 ± 0.5
	Sesquiterpene hydrocarbons (sh)				1.7 ± 0.3		2.4 ± 0.0	97.8 ± 0.7
	Oxygenated sesquiterpenes (os)					13.8 ± 5.1	0.8 ± 0.1	
	Diterpene hydrocarbons (dh)					20.4 ± 2.7		
	Apocarotenoids (ac)					1.6 ± 2.0		
	Non-terpene derivatives (nt)					8.2 ± 6.3	5.3 ± 0.7	0.3 ± 0.1
	Total Identified				100.0 ± 0.0	97.1 ± 3.0	99.8 ± 0.2	99.4 ± 0.6

^a^ Compounds were present at ≥0.1% in at least one of the analyzed essential oils. ^exp^ Linear retention index relative to *n*-alkane on the DB5 column; ^lit^ linear retention index reported by Adams, 2007; * linear retention index reported NIST 2014 [24]; ^$^ linear retention index pherobase [25]. Results are presented as the mean of three replicates ± SD.

**Table 2 molecules-26-02826-t002:** Identified compounds in the EOs of *S. aurea*, *S. dentata*, and *S. scabra.*

	Compounds ^a^	Classe	L.R.I ^exp^	L.R.I ^lit^	*S. aurea*	*S. dentata*	*S. scabra*
					Relative Percentage (%)		
1	Tricyclene	mh	927	921		0.5 ± 0.2	
2	α-Thujene	mh	930	924		0.2 ± 0.0	
3	α-Pinene	mh	939	932	1.3 ± 0.4	10.2 ± 1.9	0.8 ± 0.2
4	Camphene	mh	954	946	0.1 ± 0.1	10.0 ± 2.1	
5	3,7,7-Trimethyl-1,3,5-cycloheptatriene	nt	971	970 *	0.2 ± 0.1		
6	Sabinene	mh	975	969			0.2 ± 0.0
7	β-Pinene	mh	979	974	0.4 ± 0.1	3.2 ± 0.7	0.6 ± 0.1
8	Myrcene	mh	991	988	1.0 ± 0.3	0.5 ± 0.1	0.6 ± 0.1
9	α-Phellandrene	mh	1003	1002	0.6 ± 0.2	0.3 ± 0.1	
10	α-Terpinene	mh	1017	1014	0.1 ± 0.2	0.5 ± 0.1	
11	*p*-Cymene	mh	1025	1020	0.2 ± 0.1	0.6 ± 0.1	0.1 ± 0.1
12	Sylvestrene	mh	1028	1025	1.5 ± 0.4		
13	Limonene	mh	1029	1024	2.9 ± 0.7	2.6 ± 0.5	1.5 ± 0.2
14	δ-3-Carene	mh	1031	1021 *	7.8 ± 1.7	1.7 ± 0.3	0.2 ± 0.0
15	1,8-Cineole	om	1031	1026	2.8 ± 0.7	4.1 ± 0.9	0.4 ± 0.2
16	(*Z*)-β-Ocimene	mh	1037	1032		0.4 ± 0.1	0.1 ± 0.1
17	Benzene acetaldehyde	nt	1042	1036			0.1 ± 0.1
18	Butyl isovalerate	nt	1047	1048 *		0.2 ± 0.0	
19	(*E*)-β-Ocimene	mh	1050	1044		0.1 ± 0.1	0.3 ± 0.0
20	γ-Terpinene	mh	1060	1054	0.4 ± 0.1	1.1 ± 0.2	
21	*cis*-Sabinene hydrate	om	1070	1065		0.2 ± 0.1	
22	*p*-Mentha-2,4(8)-diene	mh	1086	1085	0.3±0.1		
23	Terpinolene	mh	1089	1086	0.3 ± 0.0	0.5 ± 0.1	0.1 ± 0.1
24	*trans*-Sabinene hydrate	om	1098	1098		0.2 ± 0.1	
25	Linalool	om	1099	1095	1.1 ± 0.3	0.2 ± 0.0	
26	Nonanal	nt	1101	1100			0.1 ± 0.1
27	*cis*-Thujone	om	1102	1101	0.3 ± 0.1		0.1 ± 0.1
28	2-Methylbutyl 2-methylbutanoate	nt	1105	1106 *		0.1 ± 0.1	
29	2-Methylbutyl isovalerate	nt	1107	1103		0.4 ± 0.1	
30	*trans*-Thujone	om	1114	1112		0.2 ± 0.0	
31	*neo*-*allo*-Ocimene	mh	1131	1128	0.1 ± 0.1		
32	Camphor	om	1146	1141	0.2 ± 0.0	23.0 ± 2.4	0.2 ± 0.1
33	Borneol	om	1169	1165		0.5 ± 0.1	
34	*p*-Mentha-1,5-dien-8-ol	om	1170	1166	0.2 ± 0.0		
35	4-Terpineol	om	1177	1174	0.2 ± 0.0	1.1 ± 0.1	
36	*p*-Cymen-8-ol	om	1183	1179	0.2 ± 0.0		
37	α-Terpineol	om	1189	1186		0.2 ± 0.0	
38	Verbenone	om	1205	1204	0.3 ± 0.1		
39	β-Cyclocitral	ac	1220	1217	0.1 ± 0.1		
40	(*Z*)-3-Hexenyl isovalerate	nt	1238	1235 *		0.1 ± 0.0	
41	Eucarvone	om	1243	1146	0.1 ± 0.1		
42	Piperitone	om	1253	1249	0.1 ± 0.1		
43	Bornyl acetate	om	1289	1284		5.4 ± 0.2	0.1 ± 0.2
44	α-Cubebene	sh	1351	1345	1.1 ± 0.0		0.1 ± 0.1
45	Isoledene	sh	1375	1374	0.2 ± 0.0		
46	α-Copaene	sh	1377	1374	2.9 ± 0.1	0.2 ± 0.0	6.5 ± 1.5
47	β-Bourbonene	sh	1388	1387			2.0 ± 1.3
48	β-Cubebene	sh	1388	1387	0.1 ± 0.0		0.5 ± 0.0
49	β-Elemene	sh	1391	1389			0.5 ± 0.3
50	(*Z*)-Jasmone	nt	1393	1392	0.2 ± 0.0		0.2 ± 0.1
51	α-Gurjunene	sh	1410	1409	2.2 ± 0.1		
52	(±)-*β*-Isocomene	sh	1412	1412 *			0.4 ± 0.1
53	β-Caryophyllene	sh	1419	1417	12.5 ± 0.4	1.1 ± 0.3	8.4 ± 1.3
54	β-Copaene	sh	1432	1430	0.5±0.0		0.5 ± 0.2
55	(+)-α-Barbatene	sh	1436	1437 *			1.5 ± 0.3
56	γ-Elemene	sh	1437	1434			0.3 ± 0.1
57	Aromadendrene	sh	1441	1439	0.2 ± 0.0		0.2 ± 0.0
58	Isogermacrene D	sh	1448	1446 ^$^			0.1 ± 0.0
59	*cis*-Muurola-3,5-diene	sh	1450	1448	0.2 ± 0.0		
60	α-Humulene	sh	1455	1452	1.7 ± 0.1	0.1 ± 0.1	3.2 ± 0.3
61	Cadina-3,5-diene	sh	1458	1454 *	0.3 ± 0.0		
62	*allo*-Aromadendrene	sh	1460	1458		0.2 ± 0.1	0.1 ± 0.1
63	*cis*-Muurola-4(14),5-diene	sh	1467	1465	0.6 ± 0.0		
64	*trans*-Cadina-1(6),4-diene	sh	1477	1475	0.5 ± 0.0		
65	γ-Muurolene	sh	1480	1478	0.8 ± 0.0		0.6 ± 0.5
66	Germacrene D	sh	1485	1484	0.5 ± 0.1		32.7 ± 4.2
67	β-Selinene	sh	1490	1489	0.4 ± 0.0		0.2 ± 0.1
68	Valencene	sh	1492	1496	1.5 ± 0.1		0.7 ± 0.0
69	*cis*-β-Guaiene	sh	1493	1492	0.4 ± 0.0		
70	Bicyclogermacrene	sh	1495	1500			0.3 ± 0.0
71	γ-Amorphene	sh	1496	1495			0.1 ± 0.1
72	Viridiflorene	sh	1497	1496		0.2 ± 0.1	
73	α-Muurolene	sh	1500	1500	1.2 ± 0.1		0.4 ± 0.4
74	Cuparene	sh	1505	1504			2.5 ± 0.1
75	(*E,E*)-α-Farnesene	sh	1508	1505			1.2 ± 0.1
76	α-Chamigrene	sh	1508	1503			0.1 ± 0.1
77	*trans*-γ-Cadinene	sh	1514	1513	4.5 ±0 .3		0.6 ± 0.0
78	δ-Cadinene	sh	1524	1522	7.8 ± 0.6	0.2 ± 0.1	2.5 ± 1.1
79	*cis*-γ-Bisabolene	sh	1534	1529			0.6 ± 0.1
80	*trans*-Cadina-1(2),4-diene	sh	1535	1537 *	0.5 ± 0.0		
81	α-Cadinene	sh	1539	1537	0.4 ± 0.0		0.1 ± 0.0
82	α-Calacorene	sh	1546	1544	0.4 ± 0.0		0.1 ± 0.2
83	Selina-3,7(11)-diene	sh	1547	1545			0.2 ± 0.0
84	Germacrene B	sh	1561	1559			7.8 ± 1.9
85	(*E*)-Nerolidol	os	1563	1561		1.6 ± 0.6	2.1 ± 0.4
86	Norbourbonone	os	1563	1561			0.2 ± 0.0
87	β-Calacorene	sh	1566	1564	0.1 ± 0.1		
88	Palustrol	os	1568	1567	0.4 ± 0.1		
89	Germacrene D-4-ol	os	1576	1574	1.0 ± 0.1		0.2 ± 0.2
90	Spathulenol	os	1578	1577			0.2 ± 0.0
91	Caryophyllene oxide	os	1583	1582	3.6 ± 0.2	0.2 ± 0.1	3.4 ± 0.6
92	Gleenol	os	1587	1586	0.1 ± 0.1		
93	Viridiflorol	os	1591	1592	0.3 ± 0.0	27.7 ± 9.0	0.7 ± 0.1
94	Salvial-4(14)-en-1-one	os	1595	1594			0.2 ± 0.0
95	Ledol	os	1599	1602	1.0 ± 0.1		
96	β-Atlantol	os	1608	1608 *			0.5 ± 0.1
97	Humulene epoxide II	os	1608	1608	0.3 ± 0.0		1.0 ± 0.1
98	Junenol	os	1617	1586	1.3 ± 0.2		0.2 ± 0.0
99	1,10-di-*epi*-Cubenol	os	1619	1618	0.6 ± 0.1		
100	Humulane-1,6-dien-3-ol	os	1619	1619 *			0.6 ± 0.4
101	1-*epi*-cubenol	os	1629	1626	3.7 ± 0.4		0.4 ± 0.0
102	γ-Eudesmol	os	1631	1630			0.2 ± 0.0
103	Caryophylla-4(12),8,(13)-dien-5-*α*-ol	os	1637	1639	0.2 ± 0.0		
104	*epi*-α-Cadinol	os	1640	1638	10.2 ± 1.0		0.8 ± 0.0
105	Cubenol	os	1642	1645			0.5 ± 0.1
106	10,10-Dimethyl-2,6-dimethylenebicyclo[7.2.0]undecan-5β-ol	os	1644	1644 *	0.6 ± 0.1		
107	α-Muurolol	os	1646	1644	0.5 ± 0.1		
108	β-Eudesmol	os	1649	1649			0.3 ± 0.1
109	α-Eudesmol	os	1653	1652			1.0 ± 0.5
110	α-Cadinol	os	1654	1652	2.2 ± 0.3		0.6 ± 0.2
111	Aromadendrene oxide-(2)	os	1678	1678	0.3 ± 0.1		0.3 ± 0.0
112	Khusimyl methyl ether	os	1680	1662 *	2.1 ± 0.3		
113	Mustakone	os	1687	1676			0.3 ± 0.1
114	6-Isopropenyl-4,8a-dimethyl-1,2,3,5,6,7,8,8a-octahydro-2-naphtalenol	os	1690	1690 *			0.2 ± 0.2
115	*ent*-Germacra-4(15),5,10(14)-trien-1β-ol	os	1695	1686 *	0.4 ± 0.1		0.9 ± 0.4
116	Shyobunol	os	1701	1688	4.7 ± 0.4		
117	Valerenol	os	1736	1736 *			0.2 ± 0.1
118	Mint sulfide	sh	1744	1740			1.6 ± 0.9
119	15-Hydroxy-*α*-muurolene	os	1777	1767			0.1 ± 0.1
120	α-Costol	os	1778	1773			0.6 ± 0.3
121	Hexahydrofarnesyl acetone	ac	1844	1845 *			0.5 ± 0.4
122	Hexadecanol	nt	1880	1874		0.1 ± 0.1	
123	Farnesyl acetone	os	1919	1913			0.1 ± 0.1
124	*epi*-13-Manool	od	2056	2059			0.1 ± 0.1
125	Pentacosane	nt	2500	2500	0.1 ± 0.2	0.1 ± 0.1	0.1 ± 0.1
	EO Yield (*w*/*w*)				1.01 ± 0.2	1.53 ± 0.4	0.10 ± 0.0
	Class of compounds				*S. aurea*	*S. dentata*	*S. scabra*
	Monoterpene hydrocarbons (mh)				17.0 ± 4.6	32.4 ± 6.6	4.5 ± 1.0
	Oxygenated monoterpenes (om)				5.5 ± 1.5	35.1 ± 3.8	0.8 ± 0.7
	Sesquiterpene hydrocarbons (sh)				41.5 ± 2.0	2.0 ± 0.7	76.6 ± 2.1
	Oxygenated sesquiterpenes (os)				33.5 ± 3.6	29.5 ± 9.7	15.8 ± 3.9
	Oxygenated diterpenes (od)						0.1 ± 0.1
	Apocarotenoids (ac)				0.1 ± 0.1		0.5 ± 0.4
	Non-terpene derivatives (nt)				0.5 ± 0.1	1.0 ± 0.1	0.5 ± 0.1
	Total identified				98.1 ± 1.9	100.0 ± 0.0	98.8 ± 2.2

^a^ Compounds present at ≥0.1% in at least one of the analyzed essential oils. ^exp^ Linear retention index relative to *n*-alkane on the DB5 column; ^lit^ linear retention index reported by Adams, 2007; * linear retention index reported NIST 2014 [24]; ^$^ linear retention index pherobase [25]. Results are presented as the mean of three replicates ± SD.

**Table 3 molecules-26-02826-t003:** Results of microdilution testing of *S. aurea* and *S. dentata* EOs on selected fungal species.

EOs	*Microsporum canis*	*Trichophyton mentagrophytes*	*Aspergillus flavus*	*Aspergillus niger*	*Fusarium solani*
Salvia aurea (*v*/*v*)	2%	2%	>5%	>5%	>5%
Salvia dentata (*v*/*v*)	0.5%	1%	>5%	>5%	>5%
Itraconazole (mg/mL)	0.125	32	16	16	−
Amphotericin B (μg/mL)	−	−	−	−	8

Standard deviation was not reported because no differences were observed between the carried-out experiments. Data are presented as means ± Standard error (*n* = 3).

**Table 4 molecules-26-02826-t004:** Results of the Kirby–Bauer and microdilution assays on the studied bacterial strains.

EOs	*Staphylococcus aureus*		*Staphylococcus pseudointermedius*		*Escherichia coli*	
	MIC	Disc (mm)	MIC	Disc (mm)	MIC	Disc (mm)
*Salvia aurea* (*v*/*v*)	>10%	0	>10%	0	>10%	0
*Salvia dentata* (*v*/*v*)	>10%	0	10%	7	>10%	0
Chloramphenicol (μg/mL)	8	19	7	20	8	20

Standard deviation was not reported because no differences were observed between the carried-out experiments. Data are presented as means ± Standard error (*n* = 3).

**Table 5 molecules-26-02826-t005:** Results of antiviral activity of *S. aurea* and *S. dentata* EOs against H1N1 influenza virus.

	Inhibition of H1N1 at Indicated EO % Concentration ^a^	
Essential Oils	0.001	0.0001
*Salvia aurea*	<10%	<10%
*Salvia dentata*	93% ± 1.3%	94% ± 1.4%

^a^ Data are presented as means ± Standard error (n = 3).

## Data Availability

Not applicable.

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
