# Peer review of "Volatiles and Antifungal-Antibacterial-Antiviral Activity of South African Salvia spp. Essential Oils Cultivated in Uniform Conditions"

_molecules, 2021, doi:10.3390/molecules26092826_

Round 1
Reviewer 1 Report
In subjected paper, the Authors presented studies upon the chemical composition and biological activity of 2 South African Salvia spp. essential oils.
The topic undertaken by the authors is interesting. Authors presented this issue in a clear way by describing it correctly in a chapter “Materials and Methods”. They also presented and discussed those results in chapter “Results and Discussion”.
Minor notes:
1) Affiliations in items 1,2,3,5 (page 1) should be given in English
2) In Tables 1 and 2, I propose to include the retention indices from the literature for comparative purposes
Author Response
Minor notes:
- Affiliations in items 1,2,3,5 (page 1) should be given in English
Action: as suggested by the reviewer, affiliations have been given in English
2) In Tables 1 and 2, I propose to include the retention indices from the literature for comparative purposes
Action: the retention indices have been added in table 2 and 3
Reviewer 2 Report
The study be Najar et al is a report on the chemical composition of volatiles and essential oils extracted from three sage species of South African origin cultivated under uniform conditions. The essential oils of S. aureaand S. dentata were further investigated for their antifungal, antibacterial, and antiviral activity. According to the authors S. dentata essential oil appeared to have promising activity against the various pathogens tested. The manuscript contains a lot of data that need to be better organized and be more concise. There are some discrepancies on the manuscript’s statements and the results presents that aquire attention. Statistical analysis of results concerning the biological activity assays is missing.
- According to the abstract “Both spontaneous emissions and essential oil (EOs) composition of three species of Salvia: aurea, and S. dentata, and S. scabra were investigated for the first time” (line 15). However, data for S. aurea are not presented in section 2.1.1. Volatile organ compound analysis. In lines 370-371 is also mentioned that “It was not possible to analyse the spontaneous emissions of S. aurea because the fresh plant was not available”. Also, in lines 465-466, the authors report “The volatile emission and the essential oil composition of three sages from South Africa were investigated and for two of them were reported here for the first time”. These inconsistencies in the text should be clarified.
- The first part of the introduction is rather long; it could be more concise.
- Please, provide Figures 1 & 2 as supplementary data.
- Table 3: Please provide explanation on the result presented in this table. What do the numbers represent? Provide the unit numbers where appropriate. What is the number of replicates of these experiments? Statistical analysis is missing. Present data as mean ± SD.
- Table 4: Table 4 labelling indicates that MIC are expressed as %. This is not correct in all cases. Please provide the appropriate units of % when it is used in the table. (v/v). What is the number of replicates of these experiments? Statistical analysis is missing. Present data as mean ± SD.
- Table 5: What is the number of replicates of these experiments? Statistical analysis is missing. Present data as mean ± SD.
- Lines 278-280: Please, clarify the meaning here by rephrasing.
- Place the information in table 6 as supplementary material.
- Thorough and careful revision of the language throughout the text is highly recommended.
Author Response
The study be Najar et al is a report on the chemical composition of volatiles and essential oils extracted from three sage species of South African origin cultivated under uniform conditions. The essential oils of S. aurea and S. dentata were further investigated for their antifungal, antibacterial, and antiviral activity. According to the authors S. dentata essential oil appeared to have promising activity against the various pathogens tested. The manuscript contains a lot of data that need to be better organized and be more concise. There are some discrepancies on the manuscript’s statements and the results presents that aquire attention. Statistical analysis of results concerning the biological activity assays is missing.
1. According to the abstract “Both spontaneous emissions and essential oil (EOs) composition of three species of Salvia: aurea, and S. dentata, and S. scabra were investigated for the first time” (line 15). However, data for S. aurea are not presented in section 2.1.1. Volatile organ compound analysis. In lines 370-371 is also mentioned that “It was not possible to analyse the spontaneous emissions of S. aurea because the fresh plant was not available”. Also, in lines 465-466, the authors report “The volatile emission and the essential oil composition of three sages from South Africa were investigated and for two of them were reported here for the first time”. These inconsistencies in the text should be clarified.
Reply and Action: Thanks to the referee to have underlined these not correct sentences. We modified all the cited sentences in the Abstract as well as in the conclusion section.
2. The first part of the introduction is rather long; it could be more concise.
Action: We re-elaborated the introduction hopefully, now it is more concise.
3. Please, provide Figures 1 & 2 as supplementary data.
Action: Done. We accepted the referee’s suggestion and shift the figures 1 & 2 in the supplementary part.
4. Table 3: Please provide explanation on the result presented in this table. What do the numbers represent? Provide the unit numbers where appropriate. What is the number of replicates of these experiments? Statistical analysis is missing. Present data as mean ± SD.
Action: Done. Please see the footnote of table 3.
5. Table 4: Table 4 labelling indicates that MIC are expressed as %. This is not correct in all cases. Please provide the appropriate units of % when it is used in the table. (v/v). What is the number of replicates of these experiments? Statistical analysis is missing. Present data as mean ± SD.
Action: we provided the appropriate units and the tests were performed in triplicate, as reported in Materials and Methods. Please see table 4.
6. Table 5: What is the number of replicates of these experiments? Statistical analysis is missing. Present data as mean ± SD.
Action: The tests were performed in triplicate, as reported in Materials and Methods. Please see table 5.
7. Lines 278-280: Please, clarify the meaning here by rephrasing.
Action: We have rephrased the sentence as suggested.
8. Place the information in table 6 as supplementary material.
Reply and Action: Since Table 6 contains the antiviral results, we think it is important that it remains in the main document.
9. Thorough and careful revision of the language throughout the text is highly recommended.
Action: Done, we revised the manuscript with the help of a mother language person.
Reviewer 3 Report
The reviewed manuscript presents a well-planned and properly conducted study on the components of the volatile fraction of the aboveground parts of three species of the genus Salvia - S. aurea, S. dentata, and S. scabra. The authors applied recognized research methods in both phytochemical and microbiological parts of the experiment. The obtained results were tabulated and typical GC chromatograms of volatile compounds were presented. However, the descriptions of acronyms used and units of concentrations/dilutions of essential oils should be completed under the tables. The quality of the presentation of GC chromatograms in my opinion could be improved. In conclusion, the work should be regarded as valuable and worthy of attention.
Author Response
The reviewed manuscript presents a well-planned and properly conducted study on the components of the volatile fraction of the aboveground parts of three species of the genus Salvia - S. aurea, S. dentata, and S. scabra. The authors applied recognized research methods in both phytochemical and microbiological parts of the experiment. The obtained results were tabulated and typical GC chromatograms of volatile compounds were presented. However, the descriptions of acronyms used and units of concentrations/dilutions of essential oils should be completed under the tables. The quality of the presentation of GC chromatograms in my opinion could be improved. In conclusion, the work should be regarded as valuable and worthy of attention.
Reply and action: Thank to the reviewer for appreciating this study. As suggested, we have added an additional description of the data in the footnote of the tables.
Round 2
Reviewer 2 Report
The authors have put considerable effort to improve the manuscript and addressed the issues raised. For matters of consistency, it should beindicated what the standard deviation is in Table 6. Is this also "0.00" as in the other tables?
I have wrongly asked during the first round of the review to move Table 6 to supplementary data and I apologise for this. I meant to say Table 1, which I still think needs to be moved to the Supplementary section.
Author Response
The authors have put considerable effort to improve the manuscript and addressed the issues raised. For matters of consistency, it should beindicated what the standard deviation is in Table 6. Is this also "0.00" as in the other tables?
Action: The Standard deviations in Table 5 (Table 6 new number in revised version) now have been added.
I have wrongly asked during the first round of the review to move Table 6 to supplementary data and I apologize for this. I meant to say Table 1, which I still think needs to be moved to the Supplementary section.
Action: Done. Please see supplementary material